# Cross-biome microbial networks reveal functional redundancy and suggest genome reduction through functional complementarity

Fernando Puente-Sánchez [1,2] ✉, Alberto Pascual-García[1], Ugo Bastolla[3], Carlos Pedrós-Alió[1] & Javier Tamames[1]

The structure of microbial communities arises from a multitude of factors, including the interactions of microorganisms with each other and with the environment. In this work, we sought to disentangle those drivers by performing a cross-study, cross-biome meta-analysis of microbial occurrence data in more than 5000 samples, applying a novel network clustering algorithm aimed to capture conditional taxa co-occurrences. We then examined the phylogenetic and functional composition of the resulting clusters, and searched for global patterns of assembly both at the community level and in the presence/absence of individual metabolic pathways.

Our analysis highlighted the prevalence of functional redundancy in microbial communities, particularly between taxa that co-occur in more than one environment, pointing to a relationship between functional redundancy and environmental adaptation. In spite of this, certain pathways were observed in fewer taxa than expected by chance, suggesting the presence of auxotrophy, and presumably cooperation among community members. This hypothetical cooperation may play a role in genome reduction, since we observed a negative relationship between the size of bacterial genomes and the size of the community they belong to.

Overall, our results suggest the microbial community assembly is driven by universal principles that operate consistently across different biomes and taxonomic groups.

Microorganisms are the second most abundant component of the global biomass on Earth[1], and the first one in terms of biodiversity[2]. In addition, they are the only ones capable of performing key ecological functions, including nitrogen fixation, methanogenesis, and all kinds of anaerobic respirations. As such, they play a critical role in driving the essential biogeochemical cycles that sustain life on our planet[3]. Microorganisms interact among themselves and with the environment, giving rise to emergent community-level properties[4,5]. These interactions are primary driving forces in microbial ecology, and determine the fate of microbial communities and, by extension, of their constituent microorganisms[4]. Therefore, the study of individual microorganisms is often not enough to predict how those very same microorganisms will behave in nature;

instead, they have to be considered in the context of the community they live in.

Microbial communities are complex and dynamic entities, and their structure arises from the interplay of four key ecological processes: selection, diversification, dispersal and drift[6,7]. Among them, selection (i.e., the existence of fitness differences between individuals) is a primary force shaping microbial community assembly[4,7,8]. Natural selection counteracts random fluctuations and acts over short timescales, which makes it experimentally tractable[9–11]. This has led to an increasing interest in synthetic microbial ecology as a tool to generate and test hypotheses regarding community assembly processes (reviewed in ref. 12). However, the simplicity inherent to synthetic microbial communities, while facilitating their precise

[1]Systems Biology Department, Centro Nacional de Biotecnología (CSIC), C/ Darwin 3, Campus de Cantoblanco, 28049 Madrid, Spain. [2]Department of Aquatic Sciences and Assessment, Swedish University for Agricultural Sciences (SLU), Lennart Hjelms väg 9, 756 51 Uppsala, Sweden. [3]Computational Biology and Bioinformatics, Centro de Biología Molecular Severo Ochoa (Universidad Autónoma de Madrid - CSIC), C/ Nicolás Cabrera 1, Campus de Cantoblanco, 28049 Madrid, Spain. ✉e-mail: fernando.puente.sanchez@slu.se

https://doi.org/10.1038/s42003-024-06616-5 **Article**

characterization, might also limit their usefulness as proxies of natural microbial communities[13,14].

A complementary approach is to study natural microbial communities and look for common assembly patterns, trying to unravel the bases of microbial association[15–19]. It has been argued that each extant community has a unique evolutionary history, which makes the search for 'laws' in Ecology futile[20]. Still, there is evidence that microbial dynamics can be generalized to a certain extent[21,22], allowing to extract useful broad principles from the study of multiple microbial communities. Such principles can be experimentally tested, improving the understanding of natural communities, and ultimately allowing to design robust synthetic communities[4,23].

In this work, we sought to identify general assembly principles by performing a cross-study, cross-biome meta-analysis of microbial occurrence data in more than 5,000 samples from ten different environments. We used a novel algorithm to create ecological assemblages from pairwise aggregations of microbial genera, which includes a statistical procedure to evaluate the significance of multi-genus assemblages. The significance is evaluated on the basis of a null model that is specific to each environmental class, attempting to separate the influence of the environment from the influence of biological interactions. This algorithm allowed the same taxa to aggregate with different partners in different habitats, thus capturing the complexity of interactions inherent to natural microbial communities. Finally, we analyzed the metabolic potential of the genera present in our ecological network in order to investigate the roles of redundancy and functional complementarity in specific metabolic pathways for microbial community assembly.

## Results

### Generation of a modular ecological network

Our taxa-assembly algorithm generates ecological networks by following the steps summarized in Fig. 1. Briefly, we collected environmental 16S rRNA gene sequences from the NCBI *env nt* database, assigned them a sample identifier and, when possible, classified them into a defined environmental hierarchy[24] (see methods). We then clustered 16S rRNA gene sequences into OTUs at the 97% level, which we subsequently classified phylogenetically[24]. For this study, we chose to classify our OTUs at the genus level. This provided a high taxonomic resolution while still allowing us to reliably combine results from different studies, which in many cases targeted different regions of the 16S rRNA gene. Thus, we obtained a database that records the presence/absence patterns of microbial genera across thousands of samples from different environments (Fig. 1a). Again, the use of presence/absence data was a necessary compromise in order to reliably aggregate data from studies that used very different methodologies. We demonstrated before the usefulness of this approach for generating cross-biome microbial association networks[25].

For this study, we focused on ten different environments: freshwater, marine water, marine sediments, hypersaline, oil, thermal, hypothermal/polar, soils, host-associated and water-treatment plants, which amounted to a total of 13,362 samples and 1424 genera in our database. After filtering (see methods), we obtained a total of 5369 samples and 966 genera for creating an agglomerative ecological network as follows. At the beginning of the process, each node represents one genus, and from the presence/absence profiles we compute all-against-all pairwise aggregation scores, which represent significant co-occurrences between pairs of genera[25] (Fig. 1b; Supplementary Note 1). The computation of the scores considers as a null model that co-occurrences occur by chance. To reduce the influence of the environment, we develop a different null model for each specific environmental subtype (see methods). We note that this does not fully eliminate biases when calculating aggregation scores, as it does not account for study-level biases (which would require the development of study-level null models that would only be possible for studies involving at least dozens of samples) or biases related to environmental differences that are too fine-grained to be captured in the microDB environmental hierarchy. We have chosen to use the term "inferred associations" throughout the manuscript to reflect the fact that, while our aim is to capture true ecological associations

between taxa, our results are nonetheless contingent to this particular combination of input data, null models, and inference algorithms.

We then iteratively cluster genera into larger environmental assemblages. At each step, we join the two nodes A and B with the highest aggregation score (Fig. 1c). A novelty of our method is that the new node A + B only conserves the samples in which both nodes are present. We then assign the remaining samples from A and from B to two new nodes A* and B*. This strategy allows investigating the aggregation of each genus with different partners in different environments, thus capturing putative conditional ecological interactions. (Fig. 1c). Finally, we recalculate the aggregation score of the nodes A + B, A* and B* with respect to all the other nodes considering the samples in which each of them is observed (Fig. 1d). We iterate this process until all pairwise scores fall below a significance threshold, obtaining a directed network that captures significant inferred associations between increasingly large groups of genera (Fig. 1e). Importantly, our procedure ensures that the whole assemblage is statistically significant.

Once the network is constructed, we use the PathoLogic algorithm[26] to predict the metabolic pathways present in the included genera and assemblages (Fig. 1f). In this way, we obtain a taxonomically and functionally annotated agglomerative ecological network that aims to represent microbial associations at different levels of complexity (Fig. 1g, Supplementary Data 1). It is important to note that the functional annotations in our network are inferred from those of the genomes of the same genera that were present in the MetaCyc database[27] and as such may not fully reflect the genomic content of the environmental strains that were originally present in our samples. To alleviate this, we have restricted our analyses to the core genome of each genera (i.e. those metabolic pathways that all present in all the available genomes of each genera), which we assume to be shared between the reference strains included in MetaCyc and the environmental strains. In Supplementary Note 2 we present a verification of these assumptions, in which we calculate the core genome of the ecologically and functionally versatile *Pseudomonas* genus using different numbers of input genomes from different habitats. Our results indicate that genus-level core genomes are reasonably similar regardless of whether they are calculated using reference genomes or metagenome assembled genomes (MAGs) coming from environmental strains, even when those genomes come from different habitats.

The final network included a total of 514 genera and 5253 samples, resulting in 1215 nodes and 1428 edges. 701 nodes corresponded to assemblages of two or more genera, with the largest assemblage having 13 members (Fig. 2a). Due to our strict cutoffs for considering an association significant, the number of genera in our assemblages is lower than the number of genera present in the input samples. Nonetheless, these assemblages represent groups of genera that are significantly associated over large numbers of samples after controlling for size and environmental biases, so they are naturally good candidates for studying putative microbial interactions.

The assemblages were distributed across the different environments, roughly following the number of input samples per environment (Fig. 2b; Supplementary Data 2). Notably, some assemblages were reconstructed in more than one environment. For example, one third of the assemblages found in marine sediments were also found in marine water, highlighting the connectivity between both environments. Conversely, the host-associated environment, while having the highest number of assemblages, shared a small fraction of them with other environments (Fig. 2b).

### Significant functional and phylogenetic redundancies in environmental microbial assemblages

Functional redundancy (i.e., the notion that multiple species can share similar roles in ecosystem functioning) has been previously reported in microbial communities, both for individual functions[5,28–31] and full metabolic reconstructions[32]. On the other hand, its generality has also been challenged by several authors[33–39]. There are several issues that complicate the quantification of functional redundancy in microbial communities. In

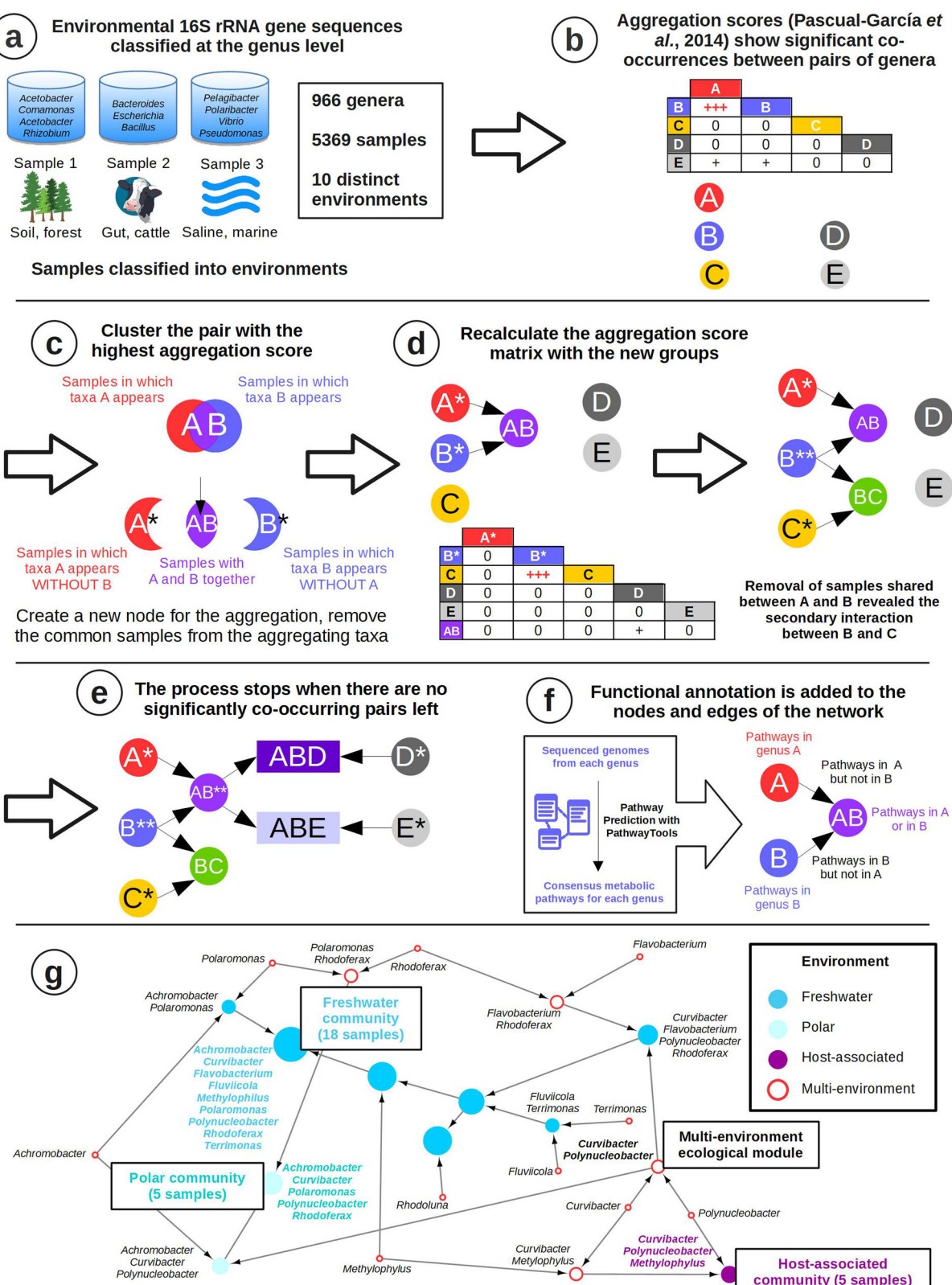

**Fig. 1 | Construction of an agglomerative ecological network. a** Database recording occurrences of genera across samples. **b** Calculation of aggregation scores showing the propensity of pairs of genera to co-occur in the samples. **c–e** Novel clustering algorithm that allows the detection of secondary interactions. **f** Functional annotation of the resulting co-occurrence network. **g** Example of the co-occurrence network generated in this work, showing a multi-environment ecological module that contributes to three different final assemblages.

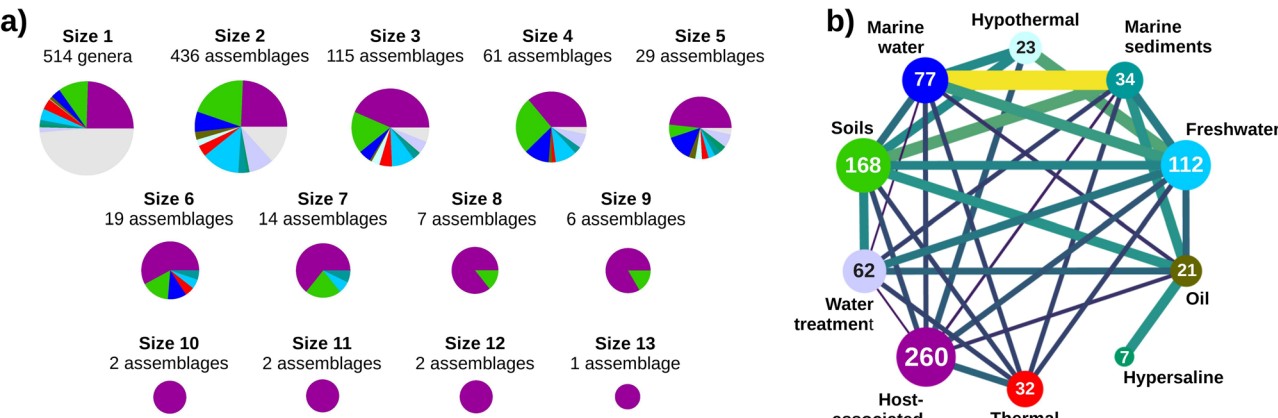

**Fig. 2 | Summary of the agglomerative ecological network. a** Number of assemblages of different sizes, and their environmental distribution. Pie chart colors indicate environments as shown in (**b**), multi-environment assemblages are indicated in gray. **b** Contribution of each environment to the network, and assemblages shared by different environments. Nodes are environments, the number of assemblages (size 2 or more) per environment is indicated inside the node. Link width and color show the assemblages that are shared between pairs of environments (as a percentage of the assemblages in the smallest environment of the pair, min 1.41%, max 29.41%). See Supplementary Data 2 for details.

microorganisms, function is often associated with phylogeny[38–40]. The presence of phylogenetically close taxa in a given community might thus increase the observed functional redundancy. Additionally, taxa have themselves different environmental preferences (e.g., host-associated vs free living, saline vs non-saline, etc[41,42].), which will aggravate this issue. Furthermore, some environments and lifestyles will favor organisms of certain genome sizes[43–46]. Since the prevalence of certain functional categories is also linked with genome size[47], selection based on genome size may indirectly enrich those functional categories, which would thus appear to be functionally redundant. Finally, gaps in the reference databases used for functional annotation can create an artefactual impression of functional redundancy as they will favor the annotation of conserved housekeeping genes (which are more likely to have close homologs in the reference database, but also to be shared between different organisms and communities) over specific genes, making the predicted functional profiles of different communities appear closer than they really are[48]. In this work, we tried to decouple function from phylogeny, the environment, and genome size, in order to provide a less biased characterization of phylogenetic and functional redundancy in environmental microbial assemblages.

We first compared the average pairwise phylogenetic and functional similarities of the microbial assemblages obtained by our approach (*environmental assemblages*) to those of random assemblages of genera (Fig. 3a, b Random). These random assemblages were functionally annotated using a similar approach as the environmental assemblages, so comparing both can offer a view of functional redundancy that is robust against annotation biases. The functional and phylogenetic distances in the environmental assemblages (blue and green boxplots in Fig. 3a, b) were significantly lower than expected by chance (Fig. 3a, b, Random vs Real, single environment), pointing to the existence of phylogenetic and functional redundancy. Furthermore, the size 2 assemblages that were detected in more than one environment (green boxplots) had a higher functional and phylogenetic redundancy than single-environment ones (blue boxplots), suggesting a relationship between functional redundancy and the ability to cope with environmental change. This result was however not significant for size 3 assemblages, perhaps due to the fact that only five size 3 assemblages were found in more than one environment.

We then aimed to control for possible confounding factors by creating random assemblages in which the genera came from the same environmental subtype, which is the most detailed environmental classification in the microDB database (differentiating for example between coastal, open and deep marine samples, see ref. 24 for details). After doing this, we further controlled the random assemblages so that their average phylogenetic similarities were the same as for the environmental assemblages (Fig. 3a, b,

Random, same environment, same phylogenetic distance). These phylogenetically-equivalent random assemblages had a higher functional redundancy (i.e., lower average distance) than completely random assemblages, which was expected since phylogenetically related organisms tend to be functionally similar[40]. Notably, while size 2 environmental assemblages had a significantly higher functional redundancy than these phylogenetically equivalent random assemblages, the effect was non-significant for higher assemblages sizes, highlighting once again the inextricable relationship between taxonomy and function.

The average number of pathways per genus (used here as a proxy for genome size) was significantly smaller in the environmental assemblages than in the random ones, even after controlling for the environment and phylogenetic relatedness (Fig. 3c). This effect was more noticeable for larger assemblages, which would be compatible with a "Black Queen"-like model of genome streamlining through public good sharing in complex communities[49] (see next section).

In order to control for the effect of average genome size, we also created random assemblages in which the average number of pathways per genus was similar to that of the environmental assemblages (Fig. 3a, b, Random, same environment, same number of pathways). Functional and phylogenetic redundancy was significantly higher in the environmental assemblages than in these genome-size-equivalent random assemblages, showcasing once again their apparent prevalence in environmental communities.

## Relationship between pathway redundancy, pathway specificity and assemblage size in environmental microbial assemblages

The results presented in the previous section obeyed to genome-wide selection patterns, but we were also interested in the selective pressures affecting individual metabolic pathways. Selection may result in pathway specificity (i.e., a pathway appearing only in one member of an assemblage), due to competitive exclusion effects (only the best competitor for a contested resource involving that pathway is present in the assemblage) or cooperative interactions (a complex route being divided among different organisms, or a common good being supplied by one member of the assemblage[49]). Conversely, a metabolic pathway will have low specificity (i.e., it will be redundant) if it is required by most or all members of a microbial assemblage, as would happen for housekeeping pathways, or for pathways selected by a common abiotic constraint in a given environment (e.g., anoxia).

In order to investigate whether individual metabolic pathways are more redundant or specific in environmental assemblages than expected by chance, we first created a consensus network by selecting the terminal assemblages (i.e. those with no outgoing edges to larger assemblages, thus representing the final product of our clustering algorithm) with high

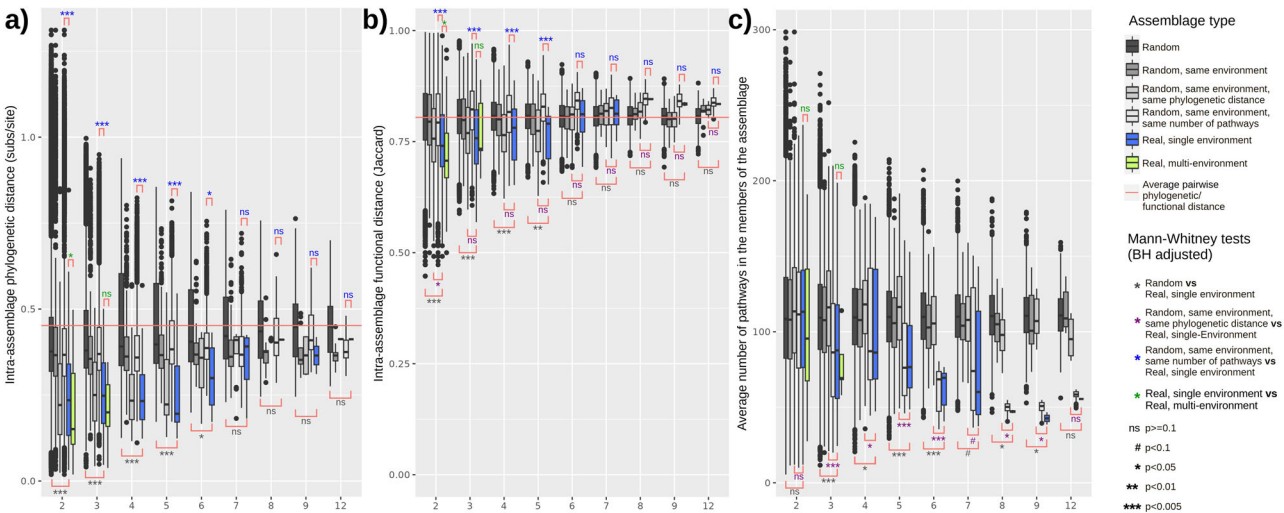

**Fig. 3 | Phylogenetic and functional redundancy in environmental versus random assemblages of microbial taxa.** Boxplots represent the distributions of average (**a**) pairwise phylogenetic distances, (**b**) pairwise functional distances and (**c**) number of pathways in the members of increasingly large assemblages (x-axis). Boxplot colour shows assemblage type: 1) fully random assemblages (dark grey), 2) environmentally-equivalent random assemblages (medium gray), in which taxa come from the same environment, 3) environmentally/phylogenetically equivalent assemblages, with taxa from the same environment and average phylogenetic distances similar to those found in environmental assemblages (grey), 4) environmentally/genome size equivalent random assemblages, with taxa from the same environment and with the same average number of pathways as the environmental assemblages (light grey), 5) environmental assemblages appearing in only one environment (blue), or 6) environmental assemblages appearing in more than one environment (green). Significant differences between different types or assemblages were evaluated with the Mann-Whitney U test and multiple testing correction was performed with the Benjamini-Hochberg method. Horizontal red lines represent the average pairwise phylogenetic or functional distance of all the genera included in our network. Boxplots contain the median (horizontal black line), with the lower and upper parts of the boxes indicating the 25th and 75th percentiles of the underlying distributions and whiskers extending 1.5 times the interquartile range from the top and bottom of the boxes.

support (meaning that the same assemblage was consistently found after 100 independent runs of our algorithm). We then computed the number of times that each metabolic pathway appeared on each of them, and compared these results to those obtained on 1000 control assemblages with the same number of taxa, randomly assembled from taxa that belonged to the same environmental subtype as the real assemblage and had similar pairwise phylogenetic distances (see Materials and Methods). Pathways whose prevalence in a real assemblage was more extreme (either higher or lower) than on 95% of the random control assemblages were subjected to further scrutiny.

We classified each metabolic pathway according to their presence in the members of the microbial assemblage in one of the three following classes. (1) Missing, if the pathway is absent from all members of the real assemblage, but present in the random ones, suggesting that it is not needed (or actually selected against) in the habitat in which the assemblage lives. (2) Specific, if it is present in at least one member of the real assemblage, but is less prevalent in the real assemblage than in the random ones. The biochemical products of specific pathways are candidate for being shared in the assemblage through cross-feeding interactions. (3) Redundant, if the pathway is present in at least two members of the real assemblage, and is more prevalent in the real assemblage than in the random ones, as expected of a capability that is useful in the given habitat and is seldom shared through cross-feeding.

The heatmap in Fig. 4a shows the distribution of redundant, specific and missing pathways in the microbial assemblages detected by our approach. A hierarchical clustering of the assemblages based on the content of redundant, specific and missing pathways showed no clear relationship with their source environment (Fig. 4a, color legend at the y-axis). This suggests that we successfully controlled for biases coming from the source environment in our analysis, and that our results obey to other, more universal causes.

The number of pathways of the three types belonging to different MetaCyc categories is shown in Fig. 4b. For most categories redundant pathways prevail, in particular for those related to energy metabolism, such as respiration and the degradation of carbohydrates, carboxylates,

nucleotides or secondary metabolites. Pathways of inorganic nutrient metabolism, and the biosynthesis of carbohydrates, lipids, amines/amides and secondary metabolites also tend to be redundant. We hypothesize that these pathways are redundant because they favor the use of the resources available in the given habitat, also consistent with the fact that the second most frequent type of these categories is "Missing". In contrast, the biosynthetic pathways for amino acids, cofactors and nucleotides/nucleosides tend to be missing or specific. These results are consistent with our interpretation of specific and redundant pathways presented above.

An interesting result, presented in Fig. 3c, is that environmental assemblages have on average smaller genomes than expected by chance, particularly if they contain many members. To further explore this observation, we show in Fig. 4c the average pathways per genome (proxy of genome size) for both small (< 5 members) and large (5+ members) assemblages, taking into consideration whether those pathways were classified as redundant, specific or missing in those assemblages (results using cutoffs of 4 and 6 members were qualitatively similar and are shown in Supplementary Fig. 1).

Small and large assemblages had a comparable number of redundant pathways after controlling for assemblage size (Fig. 4c, red). This is consistent with our previous interpretation of redundancy being caused by habitat filtering: the pathways involved in utilizing the resources (or overcoming the challenges) present in a given habitat will be enriched in its resident microorganisms, and this effect depends little on how many different microorganisms are present in those samples (i.e. whether the assemblage is small or large).

In contrast, large assemblages presented more specific pathways than small assemblages (Wilcoxon test, $p < 0.001$). A possible interpretation is that large communities offer a larger variety of "public goods" that are shared by the members of the community, and that these conditions allow reduced metabolic cost and genomic streamlining, which act as a selective force favoring the formation and maintenance of these large communities. This interpretation is consistent with recent

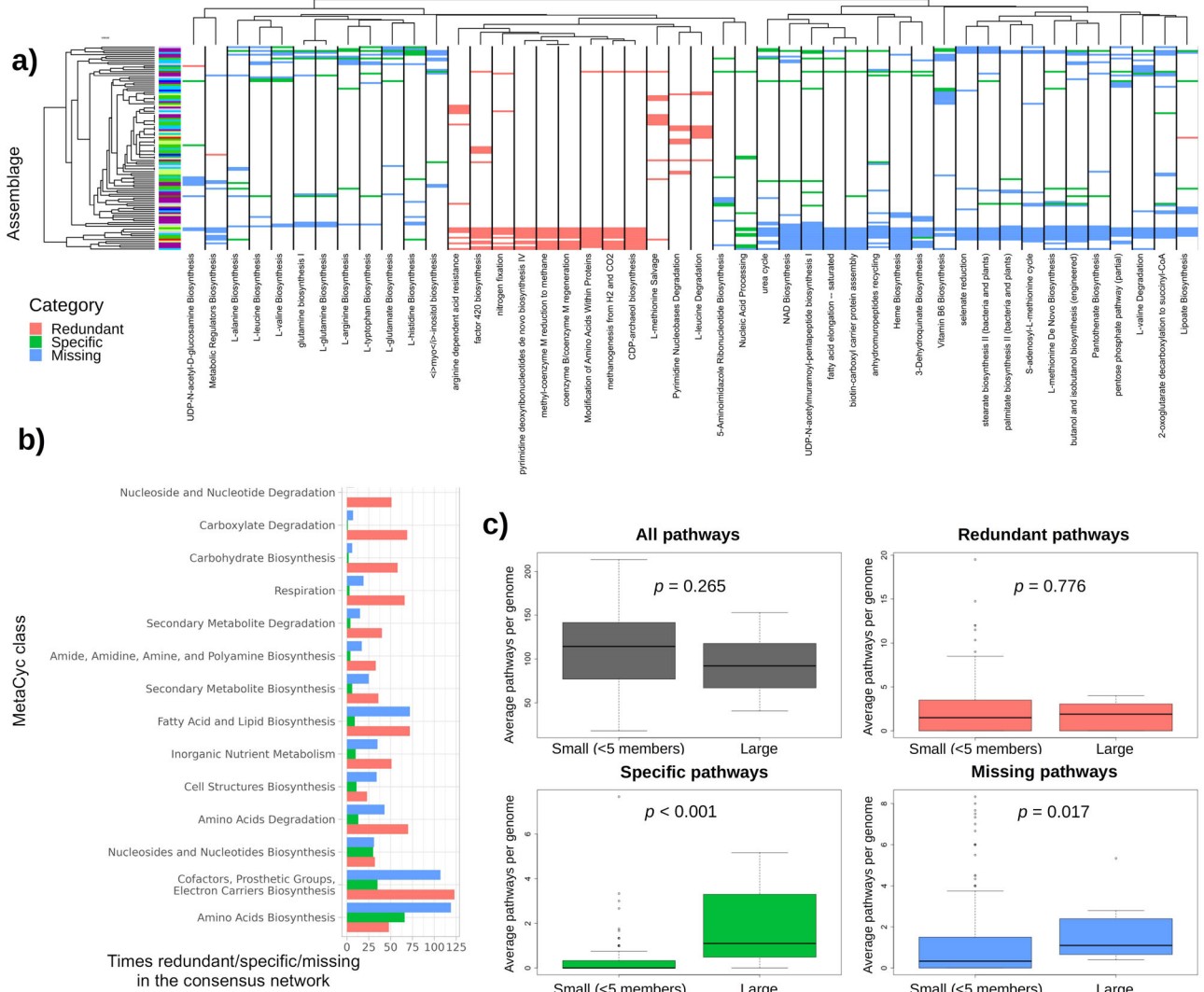

**Fig. 4 | Signs of selection in individual metabolic pathways. a** MetaCyc pathways redundant, specific and missing in the consensus network. Only pathways that were redundant, specific or missing in 10 or more assemblages are shown. The color legend in the y-axis dendrogram shows the source environment for each assemblage, following the color code shown in Fig. 2. Multi-environment communities are colored in light green. **b** Number of times each MetaCyc class was redundant, specific and missing in the consensus network. Only the 15 MetaCyc classes with the highest deviation from the random communities are shown. **c** Average pathways per genome in small (< 5 members) and large (5+ members) assemblages. First panel (grey) shows the total differences between the small and the large assemblages, the other three (red, green, blue) show the differences considering only redundant, specific, and missing pathways respectively. Boxplots contain the median (horizontal black line), with the lower and upper parts of the boxes indicating the 25th and 75th percentiles of the underlying distributions and whiskers extending 1.5 times the interquartile range from the top and bottom of the boxes.

simulation studies[50,51] and with our observations that amino acid biosynthesis is the biochemical class whose pathways are most frequently specific (see Fig. 4b) and that the fraction of assemblages in which the biosynthesis of a given amino acid is specific is significantly correlated with its biochemical cost (Fig. 5; next section).

Finally, missing pathways were also more numerous in large communities, albeit the increase was much weaker (Wilcoxon test, $p = 0.017$). According to our initial hypothesis, missing pathways would also obey to abiotic filtering (in this case being pathways that are not needed or are detrimental in a given habitat). Under this interpretation, we would expect them to be equally represented in small and large assemblages. A possible explanation for the weak enrichment in the large assemblages may be that missing pathways are pathways that are absent from the real assemblage but occur by chance in the randomly assembled communities, and their number tend to be higher in random assemblages with more members, for purely statistical reasons. Although a similar effect may also

hold for specific pathways, it is expected to be weaker than for missing pathways, supporting our biological interpretation of the difference between large and small assemblages for specific pathways.

## Patterns of amino acid auxotrophy in environmental microbial assemblages
As discussed above, environmental microbial assemblages are more functionally redundant than expected by chance (Fig. 2b). In spite of this, it is also true that certain pathways tend to appear in fewer members, particularly for larger (> = 5 members) assemblages, which we hypothesize is due to genome streamlining facilitated by the sharing of public goods. Microorganisms are well known to engage in complex interactions[22], among which auxotrophy and cross-feeding are perhaps the most studied[52]. We therefore focused on the redundancy/specificity profiles of pathways related to amino acid biosynthesis, as they are the one of the metabolites most usually involved in such processes[53].

**Fig. 5 | Average fraction of auxotrophs in large (>= 5 members) assemblages vs biosynthetic cost of amino acid biosynthesis pathways.** Blue line: linear regression model of specificity vs cost for all amino acids except for tryptophan ($R^2 = 0.55$, $p < 0.001$). Grey area: 95% confidence interval for the linear models. Red dashed line: linear regression model of specificity vs cost for all amino acids including tryptophan ($R^2 = 0.16$, $p = 0.044$). Amino acid biosynthesis costs were obtained from[55].

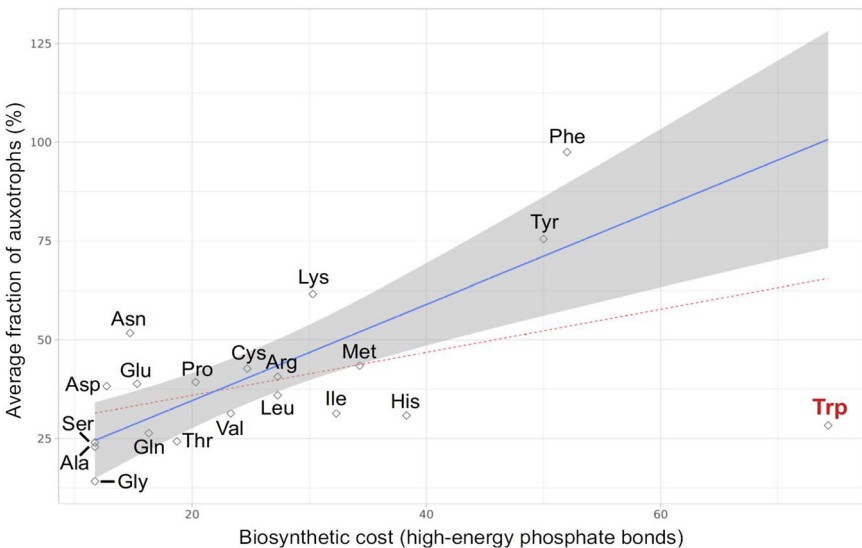

In order for auxotrophy to be a viable strategy, the potential benefits must be higher than the drawbacks derived from the resulting loss of autonomy[54]. Accordingly, the large (>= 5 members) environmental assemblages captured in our study contained more auxotrophs for expensive amino acids than for cheap ones, with the exception of tryptophan (Fig. 5, $p = 0.044$ for all amino acids, $p < 0.001$ after removing tryptophan). The comparatively lower prevalence of tryptophan auxotrophs can be explained due to its tight regulation: not only is the use of this expensive amino acid minimized across the proteome[55], but it is also seldom leaked into the environment[56,57]. The difficulty of finding free tryptophan in nature might thus partly negate the potential benefits of auxotrophy. On the other hand, since tryptophan is only required in small amounts, these benefits will be lower than otherwise suggested by its per-molecule biosynthetic cost.

These observations are consistent with those of[58], which similarly reported a larger prevalence of auxotrophy and cross-feeding for expensive amino acids. We note that this does not preclude the exchange of cheap amino acids as described in ref. 59. However, this exchange might not result in the emergence of auxotrophy, as the low cost of the exchanged metabolite might not be enough to offset the penalties associated with autonomy loss.

## Discussion

We presented a cross-study, cross-biome meta-analysis of microbial occurrence data in more than 5,000 samples from ten different environments, using a novel network generation algorithm aimed at capturing the conditional co-occurrence relationships that commonly appear in environmental microbial communities. Due to the limitations inherent to our dataset, we have purposefully restricted our analyses to relatively coarse taxonomic and functional annotations. The use of genome-resolved metagenomics on libraries generated using standardized protocols, while not exempt of biases, is a promising avenue to increase the resolution of these studies, and reveal yet unknown ecological patterns. Nonetheless, in this study we have found strong qualitative trends in microbial community assembly that appear to be conserved across different communities and biomes.

Our top-down approach complements the work already developed in synthetic communities[57–59], since it builds upon data from real environmental communities, and summarizes complex dynamics that may be difficult to replicate in experimental settings. For example, the establishment of cross-feeding interactions is expected to be subject to cost-to-benefit balance. However, the cost of the same metabolite is often context-dependent and can vary widely across microbial species and environments[60]. Microbial communities can also have different degrees of spatial structuring, which affects the range of beneficial interactions that can be established[61]. Microbial diversity is another key factor that influences

community assembly, due to its effect on stability. A diverse community will have different species that perform the same function, and this functional redundancy will make such communities more resistant to perturbations[62]. Additionally, the increased number of potential partners facilitates the establishment of weak interactions[63], which in turn allow for the development of mutualism without compromising community stability[64,65].

In spite of the wide range of ecosystems analyzed in this study, we were able to detect consistent patterns of functional redundancy and auxotrophy, hinting at the existence of conserved, biome-agnostic principles governing the assembly of microbial communities. We found that functional redundancy is ubiquitous in environmental microbial assemblages, and hypothesize that it is driven by environmental selection for some biochemical processes. We also discovered that the number of biochemical pathways per genome (which is correlated with genome size) is negatively correlated with the size of the microbial assemblages. This observation hints at interactions between members of the same assemblage, and in particular at "labor specialization", i.e. the possibility that some leaky biochemical functions possessed by some members of the assemblage are exploited by other members, allowing them to reduce their biochemical investment and their genome size. This labor specialization would generate a potential selective force behind the maintenance of large communities, as suggested by recent theoretical[50,51] and observational[66–68] studies. In agreement with this interpretation, our results suggest that, in spite of the prevalence of functional redundancy, auxotrophy commonly occurs in environmental microbial communities, particularly for costly compounds.

Overall, our results show that redundancy and auxotrophy are not mutually exclusive, but rather can coexist in microbial communities from different origins. Combining a background of functional redundancy with cooperation in the biosynthesis of key nutrients might thus be a useful design principle for engineering more robust microbial communities in the future.

## Materials And Methods
### Description of the data set

We obtained the data from the microDB database (formerly envDB[24]) following the procedure in ref. 41. The database comprises more than 20,000 environmental samples and their associated 16S rRNA gene sequences, with each sample classified at three levels: environmental supertype (e.g. aquatic), environmental type (e.g. freshwater) and environmental sybtype (e.g. river), thus informing of the presence or absence of taxa across a wide range of ecosystems. The genus level was chosen as the taxonomic working unit because it provided a good balance between the taxonomic resolution, the ability to accurately classify partial fragments of the 16S rRNA gene coming from different regions, and the

sparsity of the observations. In this study, we only considered samples coming from the following environments (as defined in the microDB classification): freshwater, marine water, marine sediments, hypersaline, oil, thermal, hypothermal/polar, soils, host-associated and water-treatment plants. In order to more reliably aggregate results from studies that used very different methodologies, data were binarized into a matrix that recorded the presence/absence of genera across samples. Samples with less than five genera and genera present in less than five samples were excluded from further analysis. This left a total of 966 genera distributed across 5369 samples from 10 environments.

## Detection of significant inferred associations between pairs of taxa

For a given pair of taxa $i$ and $j$ that co-occurr in N out of M samples, we define its aggregation score $S_{ij}$, which represents their propensity to appear together in the same samples, as the negative logarithm of the conditional probability of $i$ and $j$ co-occurring in more than N out of M samples. The original implementation of the aggregation scores can be found in[25], the implementation used in this work is detailed in Supplementary Note 1. Briefly, we used the null model from[69] that estimates the probability that a given taxon is observed in a given sample under the assumption of no interaction between taxa. We developed a different null model for each environmental subtype (the finest-grained environmental classification available in microDB). By doing this, our null models attempt to control for environment-specific biases. After inferring the parameters of the null models, we used them to generate 1000 random presence/absence matrices for each of the ten studied environments, with the same row and column totals as the real matrix. These random matrices allow to assess the influence of cosmopolitanism (i.e. the number of samples in which taxa were present) into the aggregation scores. To obtain the aggregation scores, we calculated he probability that two taxa co-occur in the number of samples observed following the algorithm in Supplementary Note 1. Aggregation scores were then transformed to Z-scores related to the mean and standard deviation of the null aggregation scores of pairs of taxa with similar cosmopolitanism. Finally, we derived a Z-score cutoff from the distribution of null Z-scores such that the False Positive Rate (i.e., the rate of significant aggregations in the null model) was not larger than 0.0001. Pairs of taxa with a Z-score higher than the cutoff were deemed significantly associated in our samples.

## Network generation

We generated an ecological network representing significant inferred associations between groups of taxa across multiple environments through the following steps:

1. For each of the ten environments included in this study:
   11. Compute aggregation Z-scores between pairs of taxa $i, j$ in samples $a$ from the binary presence/absence matrix $X_{ia}$ and the probability matrix $\pi_{ia}$ as described in Supplementary Note 1.
   12. Create 100 independent networks (in order to minimize path dependency during the clustering process) applying the following clustering procedure. We will refer generically to "nodes" for both individual taxa (e.g. elements at the beginning of the algorithm) and assemblages (taxa clustered together):
   121. While there are significantly associated pairs of nodes appearing together in more than 5 samples:
      1211. Select one significantly associated pair $i,j$ at random, weighted by its aggregation Z-score so that pairs with higher aggregation scores are more likely to be selected.
      1212. Create a new node $k$ that represents the aggregation of the selected pair of nodes $i,j$ in the samples in which they appear together, with $X_{ka} = X_{ia} \cdot X_{ja}$ and $\pi_{ka} = \pi_{ia} \cdot \pi_{ja}$.
      1213. Create the links $i \to k$ and $j \to k$.
      1214. Replace the values for $i$ and $j$ in the presence/absence matrix and in the probability matrix, so that they represent the presence of $i$

and $j$ in the samples in which they do not appear together, with $X_{i'a} = X_{ia} \cdot (1 - X_{ja})$, $\pi_{i'a} = \pi_{ia} \cdot (1 - \pi_{ja})$, $X_{j'a} = X_{ja} \cdot (1 - X_{ia})$ and $\pi_{j'a} = \pi_{ja} \cdot (1 - \pi_{ia})$.
      1215. Recalculate the aggregation Z-scores from the new X and $\pi$ matrices.
2. Combine the 1000 independent networks (100 networks from each of the 10 environments) into a single network as follows:.
   21. The combined network contains all the nodes present in the individual networks. Nodes containing the same taxa in the individual networks are collapsed into a single node in the combined network.
   22. All incoming and outgoing edges present in the individual networks are added to the collapsed nodes in the combined network.
   23. For each node and edge, we define its support value as the number of individual networks in which that node or edge was observed. Nodes and edges with a support value smaller than 10 are discarded.
   24. Nodes are annotated based on the source environment of the individual networks in which they were found.

## Environmental and bibliographic annotation of assemblages

For each sample, the microDB database contains its isolation source, as originally found in the NCBI GenBank database, as well as the Pubmed ID (PMID) of any published work related to it. We annotated each assemblage representing a significant aggregation of two or more genera with the isolation sources and related PMIDs of the samples in which the genera appeared together.

## Functional annotation of assemblages and intra-assemblage functional redundancy

We used the MetaCyc database version 19[27] to download the predicted reactomes for all the sequenced genomes from the genera included in our network (Supplementary Data 3). For each genome, we predicted its metabolic pathways from its reactome using an in-house implementation of the PathoLogic algorithm as described in[26]. As a deviation from the original algorithm, we did not add a more lenient prediction rule for energy metabolism pathways, as we found out that doing so would result in false positive predictions (e.g. sulfate respiration would be predicted for *Escherichia*). The fraction of genomes from each genus that contain each pathway is reported in Supplementary Data 4. We considered that a pathway is present in a genus if it is predicted in all of the complete genomes from that genus (i.e. the pathway belongs to the core genome of the genus). In Supplementary Note 2, we show how genus-level core genomes are reasonably similar regardless of whether they are calculated using reference genomes or genomes from environmental strains, and that as such our functional inference approach provides a good approximation to the core core genomes of the environmental strains that were originally present in our samples. We then defined the pathways present in an assemblage $\{R\}_a$ as the set union of the pathways present in its constituent genera. We also defined the average pairwise functional distance of an assemblage as the average of the the of the all-against-all Jaccard dissimilarities (1 – the Jaccard Index[70]) between the pathway vectors of its constituent genera.

## Phylogenetic distance between genera and intra-assemblage phylogenetic distances

We used 16S rRNA gene sequences from the Greengenes database[71] to obtain estimates of the phylogenetic distances between genera. First, we selected a representative full-length 16S rRNA gene sequence for each prokaryotic species in the database, usually the type strain. Then, we calculated the distance between the aligned sequences as the number of substitutions per site using RaxML with a GTRGAMMA model[72]. We calculated distances between genera as the median of the distances between the species belonging to those genera. We then calculated the average pairwise phylogenetic distance between the constituent genera of each assemblage.

https://doi.org/10.1038/s42003-024-06616-5 **Article**

### Detection of significant functional and phylogenetic redundancies at different assemblage sizes

For each assemblage size, ranging from 2 to 12 genera (the largest assemblage present in our graph for which all genera could be annotated) we compared the average functional and phylogenetic distance distributions of the assemblages present in our network to those of random assemblages of the same genera. Assemblages in which one or more genera could not be functionally annotated or lacked phylogenetic distance estimations were ignored for this and subsequent computations, leaving a total of 429 fully annotated nodes. Multi-environment assemblages (i.e. assemblages of genera that were considered significant in more than one environment during our clustering process) were treated separately from single-environment ones. For each real assemblage, we generated four different kinds of random assemblages:

a. 1000 random assemblages with the same number of genera as the real assemblage.
b. 100 environmentally-equivalent random assemblages with the same number of genera as the real assemblage, such that their genera came from the same environmental subtype (i.e. the finest environmental classification available in the microDB database, see ref. 24).
c. 100 environmentally/phylogenetically-equivalent random assemblages with the same number of genera as the real assemblage, such that their genera came from the same environmental subtype and the average pairwise phylogenetic distances in the random assemblages differed by 0.05 substitutions per position or less from the average pairwise phylogenetic distance of the original assemblage. This was done in order to assess whether functional redundancy could be explained by phylogenetic similarity and source environment alone.
d. 100 environmentally/genome size-equivalent random assemblages with the same number of genera as the real assemblage, such that their genera came from the same environmental subtype and the average number of pathways per genus differed by 20% of less from the average number of pathways in the original assemblage.

We assessed significant differences between different types or assemblages with the Mann-Whitney U test. For each metric (average intra-assemblage functional distance, average intra-assemblage phylogenetic distance and average number of pathways in the assemblage members) and assemblage size, the resulting $p$-values were corrected for multiple testing using the Benjamini-Hochberg method[73].

### Detection of redundant and specific pathways in the assemblages of our network

The procedure described above provided us with a per-assembly estimate of functional redundancy, but we were also interested in assessing functional redundancy on a per-pathway basis. For this, we first selected a subset of the network connected by highly supported (support > 70) edges. We then selected the terminal assemblages with no outgoing edges to larger assemblages, which represent the sink nodes of our clustering algorithm. For each of these assemblages, we then generated 1,000 phylogenetically and environmentally equivalent random assemblages (see previous section). In order to obtain a higher number of valid random assemblages, we increased the maximum difference in phylogenetic distances from 0.01 to 0.1 substitutions per position. Then, for each metabolic pathway, we compared its prevalence in the real assemblage with its prevalence in the random assemblages and classified it into one three categories:

1. Redundant, if it appeared in at least two members of the real assemblage, and its prevalence was higher than its prevalence in 95% of the random assemblages.
2. Specific, if appeared in the real assemblage, but its prevalence was lower than its prevalence in 95% of the random assemblages.
3. Missing, if it was missing from the real assemblage, but present in 95% of the random assemblages.

Finally, for each metabolic pathway, we computed the fraction of auxotrophs in a given assembly as 1-P/S where P is its prevalence in the assembly, and S is the size of the assembly.

### Reporting summary

Further information on research design is available in the Nature Portfolio Reporting Summary linked to this article.

## Data availability

The data used in this work are publicly available at https://github.com/fpusan/cross-biome-microbial-networks with https://doi.org/10.5281/zenodo.12770646. A SQL dump of the database used in this work can be found at https://github.com/fpusan/cross-biome-microbial-networks/tree/main/00-algorithm.

## Code availability

The code used in this work is publicly available at https://github.com/fpusan/cross-biome-microbial-networks with https://doi.org/10.5281/zenodo.12770646.

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

## Acknowledgements

AP-G was supported by the Simons Collaboration: Principles of Microbial Ecosystems (PriME), award number 542381, a Ramón y Cajal Fellowship from the Spanish Ministry of Science and Innovation (RyC2021-032424-I), by CSIC intramural project 20232AT031 and by grant PID2022-139900NA-I00 (AEI/10.13039/501100011033/ FEDER, UE). UB was supported through the grant PID2019-109041GB-C22/10.13039/501100011033 of the Spanish Agency of Research (AEI). Research at the CBMSO is facilitated by the Fundación Ramón Areces. FP-S was funded by grant CTM2016-80095-C2-1-R / NOVAMAR from the Spanish Ministerio de Economía y Competitividad, the Marie Skłodowska-Curie grant agreement No 892961 from the European Union's Horizon 2020 research and innovation programme and grant 2022-04801 from the Swedish Research Council.. This article first appeared online as a preprint with https://doi.org/10.1101/2022.09.11.507163.

## Author contributions

FP-S: Conceptualization, Methodology, Software, Validation, Formal Analysis, Investigation, Writing – Original Draft, Visualization; AP-G: Conceptualization, Methodology, Software, Validation, Writing - Review & Editing; UB: Conceptualization, Methodology, Writing - Review & Editing; CP-A: Writing - Review & Editing, Supervision, Funding acquisition; JT: Conceptualization, Resources, Data Curation, Writing - Review & Editing, Supervision, Funding acquisition, Project administration.

## Funding

## Competing interests

The authors declare no competing interests.
