## [Peer Review File · Communications Biology]

Reviewers' comments:

Reviewer #1 (Remarks to the Author):

This manuscript applies an innovative null modelling approach to study microbial interactions. While I find this approach to be very exciting, the data they apply the method to is unsuitable on several fronts. Most importantly, the data used varies in the techniques, and more crucially, in the primer regions different studies use. As primer region is known to be strongly related to detectability, these data should not be merged, especially not within the context of detecting co-occurrences. Furthermore, functional inference is extremely limited and biased by how well characterized a community is. Including different environments in the same analysis will also likely lead to erroneous conclusions: the error associated with functional inference in the human microbiome is much smaller than the error associated with the soil microbiome, for example. Below, find some other points that I think require additional attention. I want to emphasize that the network modelling approach is innovative and I encourage the authors to pursue this with better data (maybe the Earth Microbiome Project), and to forego functional inference in this context.

What are conditional ecological interactions? Aren't all interactions conditional?

L47. Selection can be biotic or abiotic

Why use Green Genes when it's so old?

L150. Unclear what an assemblage is. Also, why are assemblages so species poor?

L179: how many assemblages were discarded?

Figure 4a is not legible

Functional inference at the genus level is problematic because these traits are not all conserved at the genus level

Reviewer #2 (Remarks to the Author):

The authors present an analysis of a large microbial co-occurrence network across ~5,000 amplicon sequencing samples from diverse environments. Using a previously described aggregation score and novel environment-specific null models, the authors cluster taxa into co-occurring cliques and analyze the contained genomic pathways by mapping to reference genomes. They conclude that redundancy is common across environments, while also identifying some examples of highly cluster-specific metabolic pathways. The question of the extent and ecological impact of functional redundancy and genome reduction in microbial communities is still somewhat open in the field, so there is a need for expansive studies into those topics and related ecological concepts such as the Black Queen Hypothesis. Though the data set is expansive, the opaqueness regarding the used methodology and the somewhat coarse mapping of amplicon sequences to reference genomes

currently makes it difficult to evaluate whether the provided data and analysis support the claims made in the manuscript. Taking some additional care in showing that the analysis is robust to common biases, thresholds, and improving the mapping between OTUs and reference genomes can hopefully ameliorate those drawbacks.

Suggested major changes

The authors use their own previously published “microDB” database. The link provided in the manuscript can not be accessed, so I struggled in obtaining more information about the dataset. I did wonder a bit why the authors used a 14 year old dataset analyzed with methodology that is somewhat out of date (OTUs vs ASVs), given the wealth of amplicon and metagenomic shotgun sequencing data published in the last decade (like the Earth Microbiome Project or the Tara Oceans data) which might have been more adequate for the analysis (compare <https://doi.org/10.1186/s40168-020-00857-2> for instance). In particular, metagenomic shotgun sequencing data would have avoided the issues in mapping amplicon sequences to genomic content, as laid out below. However, I do understand that conducting the study on an entirely different data set may be out of scope here. In that case the reasons should be stated in the manuscript along with some context about other data sets that might be used in the future. At least validating the findings with some additional metagenomics data would definitely improve the analysis.

In lines 94-111 I was missing some details on how the authors determined thresholds for absence and presence for genera. Spurious low read assignments for genera and species are common in amplicon sequencing data (<https://doi.org/10.1038/s41396-021-01027-4>). Was any evidence weighting used for the presence/absence of a genus? For instance, was there a distinction between very low read and high read abundances or a sensitivity analysis to minimize false positives? In a similar vein, did the authors adjust for library size and compositional effects that can influence the false negative rate when discretizing the data?

I think it is a great idea to use environmental null models for the aggregation scores, but I wonder if there may also be study-specific biases that would require different null models. It seems like the data spans several studies using different protocols for DNA extraction and library prep that can enrich or deplete taxa present in the sample (see <https://pubmed.ncbi.nlm.nih.gov/31502536>). I think one should also check that the null models are indeed consistent/homogenous within an environment, even when estimated per study. If this is not the case, the null models should probably be estimated on a per study level.

The mapping of OTUs to genome content seems somewhat problematic to me. Even strains in the same genus will have very different genomes when comparing samples from very different environments, as was done here. Though there is some phylogenetic niche conservatism, there are

large differences in accessory genomes which often contain a multitude of enzymes. There are also biases that get introduced by the large heterogeneity in the number of used genomes for each taxon in the analysis, as shown in Table S2. I feel that a better approach would have been to map each OTU to the closest reference genome isolated from the same environment or at least to the one reference genome with the closest 16S sequence to the OTU cluster. This would represent the actual genetic content of the specific genera in their environment better than a mapping to a generic (pan-)genome of the genus.

The definition of redundancy strikes me as somewhat unusual. Usually, functional redundancy denotes genes/enzymes/pathways which are provided by several taxa in a single microbial community. However, the authors define a pathway redundant if it appears more frequently in the observed microbial communities than in random ones, even if it occurred only once. To me, this would be a generalist or essential pathway, but not a redundant one.

Suggested minor changes

The authors repeatedly call the constructed network an ecological network. However, one hallmark of ecological interactions between species or strains is that they are often asymmetric. For instance, a strain A can inhibit a strain B while strain B simultaneously promotes the growth of strain A (which would be exploitative competition, but other ecological interactions such as competition and mutualism are usually asymmetric as well). Based on what is described in the supplement, the calculation of the null probabilities and aggregation scores is commutative and can thus not capture asymmetric interactions. So I would not call those measures of ecological interactions but rather co-occurrences or correlations. So this should be delineated from other computational methods that try to quantify (asymmetric) ecological interactions (for instance <https://doi.org/10.1038/s41467-017-02090-2> or <https://doi.org/10.1038/ncomms15393>). I don't think this distinction would diminish the impact of the manuscript.

It is stated that the relationship of community assemblage size and functional pathways is supposed to be independent of the environment (lines 286-289), but the data in Figure 2 shows that pretty much all larger assemblages are specific to host-associated environments. So how was that independence insured in Figure 3 for instance?

Because, Figure 3 shows pretty much all-vs-all hypothesis testing between many groups, some correction for multiple testing would be advisable. Also, the figure legends are fairly small in Figure 3.

In Figure 4c, how was the specific threshold (<5) between small and large assemblages chosen?

Lines 311-313 would read better as “The average number of pathways per genus (used here as a proxy for genome size) was reduced in larger assemblages when compared to random assemblages [...]” (or similar without the “Regarding...”)

When discussing functional redundancy it might be worthwhile to mention the common pitfalls in the discussion (see the recent <https://doi.org/10.1038/s41564-023-01426-7> for a good summary).

Reviewer #3 (Remarks to the Author):

Review of Puente-Sanchez et al. "Cross-biome microbial networks reveal functional redundancy and suggest genome reduction through functional complementarity"

In this manuscript, Puente-Sanchez et al. describe a meta-analysis of environmental microbiome data. The goal of this meta-analysis was to identify distribution patterns of microbial assemblages across different environments and thereby help understand community assembly patterns. A primary objective was to distinguish the effects of interactions of microbial taxa with other taxa from interactions with the environment. Taxa were studied at the genus level, and metabolic pathways were inferred from genomes of each genus. After filtering, 5,369 samples from 10 environments were analyzed, comprising 966 genera. They found a degree of functional redundancy within environments that was greater than expected by chance, but also more specificity of pathway prevalence than expected by chance. They also observed smaller genome sizes (inferred from the average number of pathways per genome) in larger communities, suggesting genome reduction. Collectively this provides evidence for auxotrophy in certain microbes and cooperation within communities.

I found this an paper an interesting read. It's well-written and explores some questions that are difficult to address without a large meta-analysis like the one undertaken. This work generates hypotheses that will spur further work in the research of ecological principles underlying microbial community assembly.

My main point of feedback is there could be improved clarity in some places. In particular:

1. Abstract - There are several sections of the abstract that could be clarified:

l.22 It's unclear here why the communities are "inferred". What makes them inferred rather than measured directly?

l.23 How does assembly apply to individual metabolic pathways? How are pathways assembled? Maybe you mean the presence or absence of those pathways.

l.25 What is "they" referring to in "when they are found"?

2. Assemblages vs. communities - In the paragraph from l.385 to l.397, it seems that the terms "assemblage" and "community" are being used interchangeably. The authors need to be careful with these terms here and throughout the manuscript. By their own definitions, an assemblage is a group of taxa (genera) that co-occur and cluster together in the network; a community is all of the taxa (genera) in a given sample.

3. Data availability - I appreciate the authors putting the data and code on GitHub. Unfortunately they are poorly documented and difficult to use. The README files are nearly empty and contain little information about the code or datasets. I could not tell if the presence/absence matrices of each genus in each sample (with environment type for each sample) was included, but this would be important to include, especially if future researchers are to work further on this dataset.

Minor comments:

l.44 Replace "on" (in "live on") with "in" or "within".

l.47 Doesn't selection also require some sort of "challenge" (eg, limited resources) where differences in fitness can be expressed? If so, suggest changing "the existence of fitness" to "requiring fitness" to indicate the fitness differences are necessary but not sufficient for selection.

l.92 Maybe add to the end of this sentence/paragraph "from 10 environments".

l.99 Change "taxa" (plural) to "taxon" (singular).

l.130 Add "/" or "-" to "presence absence" (be consistent throughout manuscript).

l.148 Which NCBI database? BioSample?

l.168 Change "GreenGenes" to "Greengenes".

l.188,194 Remove spaces around dashes.

l.226 What is hierarchical about this "environmental hierarchy"? All of the figures I've seen list the 10 environments as equivalent with no structure among them.

l.229 Change "combining" to "combine".

Fig.2 Is it possible to make the pie charts larger?

l.255 Change "Pathologic" to "PathoLogic".

l.278 Suggest changing "to" to "with".

REVIEWER #1

This manuscript applies an innovative null modelling approach to study microbial interactions. While I find this approach to be very exciting, the data they apply the method to is unsuitable on several fronts. Most importantly, the data used varies in the techniques, and more crucially, in the primer regions different studies use. As primer region is known to be strongly related to detectability, these data should not be merged, especially not within the context of detecting co-occurrences.

We thank the reviewer for the thorough review and the useful comments on our work. Since this is an extensive computational study, we had to merge data that adopt different experimental set-ups, otherwise this study would simply not be possible. We recognize that the adopted primers may have an important effect on the detection of taxa. For this reason we decided to work at the genus level, since we admit that we do not have the resolution for higher taxonomic levels. At the genus level the problem of detectability should be much alleviated. In the revised version, we modified other parts of our analysis, namely functional inference, to make it fully consistent with the low resolution that the data afford us.

Please also note that the problems with the primers may cause a taxon to be undetected, i.e. false negative, but it is unlikely that it causes false positive associations. For this reason, we decided early on to eliminate from our analysis negative associations (segregations of taxa) and to focus on positive associations. We expect that we may lose many positive associations, but the ones that we disclose should be true ones. It is noteworthy that, with our strict criteria, we still obtain many significant associations and clear qualitative trends.

Furthermore, functional inference is extremely limited and biased by how well characterized a community is. Including different environments in the same analysis will also likely lead to erroneous conclusions: the error associated with functional inference in the human microbiome is much smaller than the error associated with the soil microbiome, for example. Below, find some other points that I think require additional attention. I want to emphasize that the network modelling approach is innovative and I encourage the authors to pursue this with better data (maybe the Earth Microbiome Project), and to forego functional inference in this context.

In the new version, we modified functional inference and made it more consistent with the genus level that we adopted for detecting co-occurrences. Namely, we now consider only core metabolic pathways that are found in all the available genomes of the identified genus. In this way, we certainly lose many pathways but we are highly confident that the ones that we identify exist in the community. With this new approach many more pathways will be “Missing” in any particular community, but this will also happen in the null-model communities.

The bias inherent in the fact that some communities may be better characterized than others is unfortunately unavoidable in large scale computational studies. We construct different null models in order to limit its impact:

- i) For detecting co-occurrences, we infer a different set of parameters for each subenvironment. This is one of the strengths of the dataset we are using, since the environmental classification was manually curated by one of the groups participating in this work.
- ii) To validate results derived from the functional inference, all patterns were validated with a battery of tests in which each observed assemblage was contrasted against random assemblages of the same size and a) taxa observed in the same subenvironment, b) a similar phylogenetic relatedness, c) a combination of a) and b). Therefore, any pattern that is

significant in, say, human host environments has been contrasted with taxa also observed in that environment, preventing spurious biases.

Therefore, we expect that the null models considered make our results robust with respect to possible experimental biases that characterize each sub-environment. For instance, as the reviewer remarks, more paths are known in the human microbiome, but this fact is taken into account by the null model for functional inference.

As also pointed out by another reviewer, the alternative to functional inference would have been to derive the functions associated to each genus on each sample directly from metagenomics data, by assembling, binning as annotating high-quality Metagenome Assembled Genomes (MAGs). This would be problematic for different reasons, as MAGs are in many cases incomplete (and crucially less complete than predicted by tools such as CheckM, see <https://doi.org/10.1128/AEM.02593-20>), and their detectability is also subjected to different sources of bias related to both binning and shotgun metagenomics in general (<https://doi.org/10.1093/gigascience/giaa008>; <https://doi.org/10.1128/AEM.02593-20>; <https://doi.org/10.1101/2023.05.02.539054>). Considering this, and the large scale of our study, we chose to rely on functional inference, which is a broadly accepted practice within our field (e.g. the PICRUST tool for functional inference has near 3000 citations at the time).

Nonetheless, we shared the reviewer's concerns on the validity of functional inference in our context, and have performed extra analyses to address them. Firstly, as stated above, we have repeated all the analyses using the core genome for each genus, which we expect to be more robust against biases. We then tested these expectations by downloading more than 14,000 genome annotations from the highly versatile *Pseudomonas* genus, and comparing the core genome estimates obtained using isolate genomes (similar to the functional inference strategy used in our work) and MAGs (assumed to be free of the biases related to functional inference). These annotations were obtained from the proGenomes3 database (<https://doi.org/10.1093/nar/gkac1078>), which crucially assigns a source habitat to each genome, thus allowing to assess the effect of habitat in functional inference as suggested by the reviewer.

Our results indicate that genus-level core estimates are fairly similar regardless of whether they come from MAGs and isolate genomes, with c.a. 90% of the functions in MAG cores being present in the isolate cores. This similarity remained even when considering MAGs from different habitats, or using a small number of genomes to calculate the cores. Overall, we believe that our results sufficiently prove that functional inference based on genus-level core genomes, as employed in our work, provide a reasonably unbiased representation of the true core of the genus, even for a genus as environmentally and functionally versatile as *Pseudomonas*.

We now discuss the issues raised by the reviewer explicitly in the main text, and present our validation analyses and results in a new 9-page length Supplementary Note (Supplementary Note S2).

What are conditional ecological interactions? Aren't all interactions conditional?

They are, but traditional clustering algorithms will allow a given taxon to be part of only one cluster, that will capture either the strongest interactions for that taxon, or those occurring across the largest number of samples. Those algorithms thus fail to detect conditional interactions (i.e. the same taxon potentially having completely different partners under different conditions). Our algorithm was explicitly designed to detect such cases.

We have clarified this when describing our algorithm

“[...] This strategy allows investigating the aggregation of each genus with different partners in different environments, thus capturing putative conditional ecological interactions”

L47. Selection can be biotic or abiotic

We of course agree with the reviewer but think that our original wording in lines 45-49 was correct.

“Microbial communities are complex and dynamic entities, and their structure arises from the interplay of four key ecological processes: selection, diversification, dispersal and drift (Vellend, 2010; Nemergut et al., 2013). Among them, selection (i.e., the existence of fitness differences between individuals) is a primary force shaping microbial community assembly (Nemergut et al., 2013; Konopka et al., 2015; Louca et al., 2017).”

This definition of selection applies to both biotic and abiotic selection, which can be understood as two facets of the same overarching process (see e.g. 2010 Vellend’s conceptual synthesis). While later in the manuscript we make distinctions between the two, we don’t think this is needed at this stage in the introduction.

Why use Green Genes when it’s so old?

We did this for consistency with our previous study. We don’t think this will have a large effect since we are working at the genus level, and only 26 out of the 514 genera in our final network were absent from our GreenGenes dataset used for phylogenetic distance estimation.

L150. Unclear what an assemblage is. Also, why are assemblages so species poor?

We thank the Reviewer for raising this to our attention. We use the word assemblage to indicate the groups of significantly associated taxa that result from our clustering algorithm. Since we work at the genus level, and since the rules that we impose for considering an association significant are strict, the number of taxa in the assemblages that we derive is typically much lower than the number of species in an observed community. We now explain these important points more clearly in the revised version.

“Due to our strict cutoffs for considering an association significant, the number of genera in our assemblages is lower than the number of genera present in the input samples. Nonetheless, these assemblages represent groups of genera that are significantly associated over large numbers of samples after controlling for size and environmental biases, so they are naturally good candidates for studying putative microbial interactions.”

L179: how many assemblages were discarded?

Thanks for pointing this out. We have added this information in the text

“Assemblages in which one or more genera could not be functionally annotated or lacked phylogenetic distance estimations were ignored for this and subsequent computations, leaving a total of 429 fully annotated nodes”

Figure 4a is not legible

We apologize for this. We have increased font size and increased resolution in DPI. After this, we have double checked and the pathway names are now legible.

Functional inference at the genus level is problematic because these traits are not all conserved at the genus level

We thank the Reviewer for this important comment, which inspired us to consider for functional inference only the core genes that are present in all species of the genus. As we mention above, in this way our assembled communities contain many fewer genes. However, this same feature is present also in null-model communities, making the detection of significant Missing paths even more difficult. Despite this important modification of the procedure, all qualitative results are maintained with the new approach, which we have also validated using MAGs and isolate genomes from the generalist *Pseudomonas* genus.

REVIEWER #2

The authors present an analysis of a large microbial co-occurrence network across ~5,000 amplicon sequencing samples from diverse environments. Using a previously described aggregation score and novel environment-specific null models, the authors cluster taxa into co-occurring cliques and analyze the contained genomic pathways by mapping to reference genomes. They conclude that redundancy is common across environments, while also identifying some examples of highly cluster-specific metabolic pathways. The question of the extent and ecological impact of functional redundancy and genome reduction in microbial communities is still somewhat open in the field, so there is a need for expansive studies into those topics and related ecological concepts such as the Black Queen Hypothesis.

We thank the Reviewer for their thorough review of our paper and the positive comments on our work.

Though the data set is expansive, the opaqueness regarding the used methodology and the somewhat coarse mapping of amplicon sequences to reference genomes currently makes it difficult to evaluate whether the provided data and analysis support the claims made in the manuscript. Taking some additional care in showing that the analysis is robust to common biases, thresholds, and improving the mapping between OTUs and reference genomes can hopefully ameliorate those drawbacks.

We thank the reviewer for the interesting suggestions. We addressed all of them, as explained below in the detailed point by point response, and we think that our work has considerably improved.

Suggested major changes

The authors use their own previously published “microDB” database. The link provided in the manuscript can not be accessed, so I struggled in obtaining more information about the dataset. I did wonder a bit why the authors used a 14 year old dataset analyzed with methodology that is somewhat out of date (OTUs vs ASVs), given the wealth of amplicon and metagenomic shotgun sequencing data published in the last decade (like the Earth Microbiome Project or the Tara Oceans data) which might have been more adequate for the analysis (compare <https://doi.org/10.1186/s40168-020-00857-2> for instance). In particular, metagenomic shotgun sequencing data would have avoided the issues in mapping amplicon sequences to genomic content, as laid out below. However, I do understand that conducting the study on an entirely different data set may be out of scope here. In that case the reasons should be stated in the manuscript along with some context about other data sets that might be used in the future. At least validating the findings with some additional metagenomics data would definitely improve the analysis.

We thank the reviewer for these comments and suggestions. Indeed, repeating the study

with metagenomic data is outside our possibilities. We used microdb because it provides an accurate environmental classification that is absolutely necessary for our study. The main reason is that we need to control as much as possible for environmental information, in order to reduce its effect in the inference of ecological associations, and the environmental information contained in this db was manually curated by one of the groups participating in this work.

We apologize for the link no longer being accessible and thank the reviewer for noticing this, the server hosting the database did indeed fail. However we would like to note that a SQL dump of the database as it was used to perform this study was already available in the GitHub repository containing the data and code used to produce the manuscript, so the server being down has no effect on the reproducibility of our work.

As we wrote to Reviewer 1, who made similar comments, we agree that our study has several limitations but we think that the results are valid provided these limitations are recognized. In particular, undetected taxa may limit the number of observed co-occurrences, but we are confident that those that we identify do exist. For our large-scale approach it is necessary to work at the genus level, and it may be difficult to identify the genus of rare taxa. Moreover, it is unlikely that these rare taxa that appear in metagenomic sets can be detected in significant co-occurrence with other taxa, therefore in practice we expect that metagenomic data would not add much to our assembled communities.

(The paragraphs below come from our previous answer to a similar concern by Reviewer 1, we also include them here for convenience, as they also address this reviewer's concerns regarding the validity of functional inference and the need for additional validation using metagenomics data).

Furthermore, metagenomic data is not exempt from biases of its own. Metagenome Assembled Genomes (MAGs), which we would need to use in order to link genera to reasonably complete metabolic reconstructions, are in many cases incomplete (and crucially less complete than predicted by tools such as CheckM, see <https://doi.org/10.1128/AEM.02593-20>), and their detectability is also subjected to different sources of bias related to both binning and shotgun metagenomics in general (<https://doi.org/10.1093/gigascience/giaa008>; <https://doi.org/10.1128/AEM.02593-20>; <https://doi.org/10.1101/2023.05.02.539054>). Considering this, and the large scale of our study, we chose to rely on functional inference, which is a broadly accepted practice within our field (e.g. the PICRUST tool for functional inference has near 3000 citations at the time).

Nonetheless, we shared the reviewer's concerns on the validity of functional inference in our context, and the need for further validation using metagenomics data, and have performed extra analyses to address them. Firstly, as stated above, we have repeated all the analyses using the core genome for each genus, which we expect to be more robust against biases. We then tested these expectations by downloading more than 14,000 genome annotations from the highly versatile *Pseudomonas* genus, and comparing the core genome estimates obtained using isolate genomes (similar to the functional inference strategy used in our work) and MAGs (assumed to be free of the biases related to functional inference). These annotations were obtained from the proGenomes3 database (<https://doi.org/10.1093/nar/gkac1078>), which crucially assigns a source habitat to each genome, thus allowing to assess the effect of habitat in functional inference as suggested by the reviewer.

Our results indicate that genus-level core estimates are fairly similar regardless of whether they come from MAGs and isolate genomes, with c.a. 90% of the functions in MAG cores being present in the isolate cores. This similarity remained even when considering MAGs from different habitats, or using a small number of genomes to calculate the cores. Overall,

we believe that our results sufficiently prove that functional inference based on genus-level core genomes, as employed in our work, provide a reasonably unbiased representation of the true core of the genus, even for a genus as environmentally and functionally versatile as *Pseudomonas*.

We now discuss the issues raised by the reviewer explicitly in the main text, and present our validation analyses and results in a new 9-page length Supplementary Note (Supplementary Note S2).

As requested by the reviewer, we also mention openly in the discussion the shortcomings of our approach, and how some of them could potentially be alleviated by the use of metagenomics.

Due to the limitations inherent to our dataset, we have purposefully restricted our analyses to relatively coarse taxonomic and functional annotations. The use of genome-resolved metagenomics on libraries generated using standardized protocols, while not exempt of biases, is a promising avenue to increase the resolution of these studies, and reveal yet unknown ecological patterns.

In lines 94-111 I was missing some details on how the authors determined thresholds for absence and presence for genera. Spurious low read assignments for genera and species are common in amplicon sequencing data (<https://doi.org/10.1038/s41396-021-01027-4>). Was any evidence weighting used for the presence/absence of a genus? For instance, was there a distinction between very low read and high read abundances or a sensitivity analysis to minimize false positives? In a similar vein, did the authors adjust for library size and compositional effects that can influence the false negative rate when discretizing the data?

We work at the genus level, which should alleviate spurious taxonomic assignments (as more sequencing errors are needed to produce an incorrect genus-level classification than when working with species / 3% OTUs / ASVs). Furthermore, microDB contains a comparatively smaller number of sequences per sample since the OTUs deposited in GenBank at the time came largely from Sanger studies with low sequencing depth. As such, the number of genera per sample in our database is comparatively low, with the 90% of our samples having 31 genera or less. This also reduces the chance of spurious low-abundance assignments having a large influence when discretizing our data. Finally, we use per-subenvironment null models based on our input presence-absence matrices in order to deem whether associations are truly significant. This would further minimize the risk of spurious low-abundance assignments (if present) resulting in a spurious aggregation being reported as significant in our network.

I think it is a great idea to use environmental null models for the aggregation scores, but I wonder if there may also be study-specific biases that would require different null models. It seems like the data spans several studies using different protocols for DNA extraction and library prep that can enrich or deplete taxa present in the sample (see <https://pubmed.ncbi.nlm.nih.gov/31502536>). I think one should also check that the null models are indeed consistent/homogenous within an environment, even when estimated per study. If this is not the case, the null models should probably be estimated on a per study level.

We thank the reviewer for the comment. We actually built one model per sub-environment (the finest-grained environmental classification available in microdb) rather than per environment (e.g. the marine water environment can be divided into coastal waters, open waters, deep waters, etc...) in an attempt to control for biases at the highest resolution available to us. This was not well reflected in the original text, but only in the code we submitted to github and linked in the paper. We now make this clear in the methods section.

For building the null model of a specific sub-environment, we need a large number of samples that are classified as belonging to the same sub-environment. Given the large variability of experimental techniques, with different DNA extraction protocols and library preparation, it is not easy to classify different studies into classes of experimental techniques, and it is unlikely that these classes would be large enough to determine experiment-specific null models, except perhaps for a few cases. Moreover, on top of this we would have to impose our environmental classification.

It is not possible to derive a null model for a single study since we need a large number of samples. Therefore, the question of the robustness of the null model across different types of study is unfortunately very difficult to address. So far, we should acknowledge that the associations inferred may not be free of environmental preferences and care should be taken in interpreting them as ecological interactions. We revised our MS to ensure we consistently refer to them as “inferred associations” to emphasize that they are the product of our algorithm and null models and may not be fully free of certain biases.

We also discuss this explicitly when presenting our clustering algorithm:

To reduce the influence of the environment, we develop a different null model for each specific environmental subtype (see methods). We note that this does not fully eliminate biases when calculating aggregation scores, as it does not account for study-level biases (which would require the development of study-level null models that would only be possible for studies involving at least dozens of samples) or biases related to environmental differences that are too fine-grained to be captured in the microDB environmental hierarchy. We have chosen to use the term “inferred associations” throughout the manuscript to reflect the fact that, while our aim is to capture true ecological ecological associations between taxa, our results are nonetheless contingent to this particular combination of input data, null models, and inference algorithms.

The mapping of OTUs to genome content seems somewhat problematic to me. Even strains in the same genus will have very different genomes when comparing samples from very different environments, as was done here. Though there is some phylogenetic niche conservatism, there are large differences in accessory genomes which often contain a multitude of enzymes. There are also biases that get introduced by the large heterogeneity in the number of used genomes for each taxon in the analysis, as shown in Table S2. I feel that a better approach would have been to map each OTU to the closest reference genome isolated from the same environment or at least to the one reference genome with the closest 16S sequence to the OTU cluster. This would represent the actual genetic content of the specific genera in their environment better than a mapping to a generic (pan-)genome of the genus.

We agree with the Reviewer, and for this reason we decided to repeat the analysis considering only the core genome of each genus (i.e. the pathways appearing in all of its genomes), both in our assembled communities and in null model assemblages. We thank the Reviewer for this remark. Despite this big change, we still identify significant results, and the qualitative results are maintained.

The definition of redundancy strikes me as somewhat unusual. Usually, functional redundancy denotes genes/enzymes/pathways which are provided by several taxa in a single microbial community. However, the authors define a pathway redundant if it appears more frequently in the observed microbial communities than in random ones, even if it occurred only once. To me, this would be a generalist or essential pathway, but not a redundant one.

We thank the reviewer, we did not appreciate that with our definition a path could be considered redundant even if it is present in only one taxon of the community. We changed the definition so that now a pathway that occurs more than expected but is present in only one taxon is no longer considered. We thank the Reviewer for pointing this out. There were very few pathways that we previously considered redundant but were present in only one taxon. The qualitative results did not change.

Suggested minor changes

The authors repeatedly call the constructed network an ecological network. However, one hallmark of ecological interactions between species or strains is that they are often asymmetric. For instance, a strain A can inhibit a strain B while strain B simultaneously promotes the growth of strain A (which would be exploitative competition, but other ecological interactions such as competition and mutualism are usually asymmetric as well). Based on what is described in the supplement, the calculation of the null probabilities and aggregation scores is commutative and can thus not capture asymmetric interactions. So I would not call those measures of ecological interactions but rather co-occurrences or correlations. So this should be delineated from other computational methods that try to quantify (asymmetric) ecological interactions (for instance <https://doi.org/10.1038/s41467-017-02090-2> or <https://doi.org/10.1038/ncomms15393>). I don't think this distinction would diminish the impact of the manuscript.

We agree with the Reviewer and we changed most instances of "ecological interactions" to "co-occurrences" or "aggregations", which is the term that we used in the paper where three of us introduced the null-model approach. The term ecological interaction is also improper because we cannot be sure that significant co-occurrences come from ecological interactions rather than from habitat filtering, which is the reason why we put great care in developing environment-specific null models. However, we think that many of the aggregations that we identify originate from ecological interactions and that, although our method cannot detect asymmetric interactions, it does suggest the sign of the global effect of the interaction on the two species.

It is stated that the relationship of community assemblage size and functional pathways is supposed to be independent of the environment (lines 286-289), but the data in Figure 2 shows that pretty much all larger assemblages are specific to host-associated environments. So how was that independence insured in Figure 3 for instance?

Indeed it is true that most large assemblages tend to occur in host-associated environments. Since our null model is environment-specific and it controls for the number of taxa observed in the environment, we think that this is not an artifact but it is a real result that suggests that the host-associated environments present many significant aggregations possibly originated from mutually beneficial interactions. The sentence in lines 286-289 of the original version was

In this work, we tried to decouple function from phylogeny, the environment, and genome size, in order to provide an unbiased characterization of phylogenetic and functional redundancy in environmental microbial assemblages.

and referred directly to our use of null models that take the source environment into consideration in order to ensure that the trends reported in our work are environment-independent, to the extent of what can be done with our environmental classification. These null models are discussed in more detail in the paragraphs directly following that sentence.

Our null models of functional pathways are specific for each environment and number of taxa, therefore the null model tries to control for possible biases inherent to a specific environment. Furthermore from Figure 2 one can see that almost half of the assemblages with 5 taxa or more are not in the host-associated environment, therefore results for these large assemblages are not only specific to host-associated environments. Similarly, in Figure 4a it can be seen that the terminal assemblages used for assessing the trends in individual pathways are not particularly dominated by host-associated environments.

We have however removed the word “unbiased” from that sentence, since while we have made great efforts to minimize the impact of different biases in our analyses, we can not claim that we are fully free of them.

Because, Figure 3 shows pretty much all-vs-all hypothesis testing between many groups, some correction for multiple testing would be advisable. Also, the figure legends are fairly small in Figure 3.

We agree with the reviewer in the need for multiple testing corrections, and we thank them for their comment. We would like to point out that we are not performing all-to-all comparisons. For each assemblage size, we compare the real assemblages with different null models (and with multi-environment assemblages in a few cases) in order to test different hypotheses. Crucially, we never attempt to compare the different null models with each other. The comparisons being performed are indicated in the legend for Figure 3 (we have increased the font size as requested by the reviewer) and there it can be seen that we perform at most 4 comparisons per assemblage size. Considering this, we now report adjusted p values obtained after multiple testing correction using the Benjamini-Hochberg method. We have updated the methods and figure legend in order to reflect this.

In Figure 4c, how was the specific threshold (<5) between small and large assemblages chosen?

From Figure 2 one can see that, as noted by the Reviewer, assemblages with seven or more taxa are prevalently of the host-associated type, which undermines their generality. On the other hand, the number of assemblages with one or two taxa is very large, so that this condition is not restrictive. We are left with four to six taxa as the threshold between small and large communities. We have repeated the analysis using 4 and 6 as alternative cutoffs, and the qualitative results are the same. This is now explicitly mentioned in the text.

Lines 311-313 would read better as “The average number of pathways per genus (used here as a proxy for genome size) was reduced in larger assemblages when compared to random assemblages [...]” (or similar without the “Regarding...”)

We thank the Reviewer for the suggestion, we changed the text accordingly.

When discussing functional redundancy it might be worthwhile to mention the common pitfalls in the discussion (see the recent <https://doi.org/10.1038/s41564-023-01426-7> for a good summary).

We thank the reviewer for this valuable reference, which we promptly incorporated into our discussion. That paper makes a very good point on why the perceived functional redundancy reflects in many cases reference biases instead, as per the quote

“This claim derives from studies showing that, whereas the taxonomic composition of human metagenomes can vary hugely, functional gene prediction profiles remain remarkably

consistent. We contend that this is at least partly artefactual, as these functional comparisons are typically carried out after discarding the large proportion of metagenomic data that does not map to reference databases”

However we would like to note that we use null communities(which contain the same reference biases as the real communities) in order to assess redundancy and specificity. Thus, our claim is not simply “microbial communities are functionally redundant”, but “microbial communities are more functionally redundant than random associations of taxa”. We believe that our use of random null communities lets us avoid the common pitfalls brought up by the reviewer, and that our study (with all its caveats) has captured some real underlying patterns in how microbial communities are assembled.

REVIEWER #3

Review of Puente-Sanchez et al. "Cross-biome microbial networks reveal functional redundancy and suggest genome reduction through functional complementarity"

In this manuscript, Puente-Sanchez et al. describe a meta-analysis of environmental microbiome data. The goal of this meta-analysis was to identify distribution patterns of microbial assemblages across different environments and thereby help understand community assembly patterns. A primary objective was to distinguish the effects of interactions of microbial taxa with other taxa from interactions with the environment. Taxa were studied at the genus level, and metabolic pathways were inferred from genomes of each genus. After filtering, 5,369 samples from 10 environments were analyzed, comprising 966 genera. They found a degree of functional redundancy within environments that was greater than expected by chance, but also more specificity of pathway prevalence than expected by chance. They also observed smaller genome sizes (inferred from the average number of pathways per genome) in larger communities, suggesting genome reduction. Collectively this provides evidence for auxotrophy in certain microbes and cooperation within communities.

I found this an paper an interesting read. It's well-written and explores some questions that are difficult to address without a large meta-analysis like the one undertaken. This work generates hypotheses that will spur further work in the research of ecological principles underlying microbial community assembly.

We thank the Reviewer for the thorough review of our paper, the nice comments and the useful suggestions.

My main point of feedback is there could be improved clarity in some places. In particular:

1. Abstract - There are several sections of the abstract that could be clarified:

I.22 It's unclear here why the communities are "inferred". What makes them inferred rather than measured directly?

By inferred we mean assembled through our clustering algorithm, which is specific for each sub-environment. We agree that the term inferred is not very precise and we changed it to avoid possible confusion.

I.23 How does assembly apply to individual metabolic pathways? How are pathways assembled? Maybe you mean the presence or absence of those pathways.

Indeed we mean the presence/absence of those pathways. We now clarified this in the text.

I.25 What is "they" referring to in "when they are found"?

It referred to the communities, but it was indeed unclear. We have changed the sentence to "Our analysis highlighted the prevalence of functional redundancy in microbial communities, particularly between taxa that co-occur in more than one environment".

2. Assemblages vs. communities - In the paragraph from I.385 to I.397, it seems that the terms "assemblage" and "community" are being used interchangeably. The authors need to be careful with these terms here and throughout the manuscript. By their own definitions, an assemblage a group of taxa (genera) that co-occur and cluster together in the network; a community is all of the taxa (genera) in a given sample.

We thank the Reviewer for the suggestion. We now use the term community only to indicate the taxa in a given sample, as suggested by the Reviewer.

3. Data availability - I appreciate the authors putting the data and code on GitHub. Unfortunately they are poorly documented and difficult to use. The README files are nearly empty and contain little information about the code or datasets. I could not tell if the presence/absence matrices of each genus in each sample (with environment type for each sample) was included, but this would be important to include, especially if future researchers are to work further on this dataset.

We apologize for this, it is true that the repository was poorly documented. The presence/absence matrices for each environment were indeed included, as well as a mySQL dump of the original database used in this work and the annotated networks in different formats. We also had the auxiliary data and the exact commands used to replicate the different steps of the analysis, but this was of course not evident without further information. We have now improved the documentation of the code and datasets, describing each of the main code and results files.

Minor comments:

I.44 Replace "on" (in "live on") with "in" or "within".

Done.

I.47 Doesn't selection also require some sort of "challenge" (eg, limited resources) where differences in fitness can be expressed? If so, suggest changing "the existence of fitness" to "requiring fitness" to indicate the fitness differences are necessary but not sufficient for selection.

I.92 Maybe add to the end of this sentence/paragraph "from 10 environments".

Done.

I.99 Change "taxa" (plural) to "taxon" (singular).

Done.

I.130 Add "/" or "-" to "presence absence" (be consistent throughout manuscript).

Done.

I.148 Which NCBI database? BioSample?

It's GenBank, we now mention this in the text.

I.168 Change "GreenGenes" to "Greengenes".

Done.

I.188,194 Remove spaces around dashes.

Done.

I.229 Change "combining" to "combine".

Done.

Fig.2 Is it possible to make the pie charts larger?

Done.

I.255 Change "Pathologic" to "PathoLogic".

Done.

I.278 Suggest changing "to" to "with".

Done. We thank the Reviewer for all these detailed and useful suggestions, we changed the text accordingly.

I.226 What is hierarchical about this "environmental hierarchy"? All of the figures I've seen list the 10 environments as equivalent with no structure among them.

The environmental classification adopted by microDB is hierarchical, in the sense that it classifies samples into environmental subtypes and these into environmental types and environmental supertypes. Although we present the results at the environment level, we constructed the null models at the environmental subtype level, which is the finest grained classification available. This is now explained more clearly in the methods of the revised version.

"The database comprises more than 20,000 environmental samples and their associated 16S rRNA gene sequences, with each sample classified at three environmental levels: supertype (e.g. aquatic), type (e.g. freshwater) and subtypes (e.g. river), thus informing of the presence or absence of taxa across a wide range of ecosystems."

"We developed a different null model for each environmental subtype (the finest-grained environmental classification available in microDB). By doing this, our null models attempt to control for environment-specific biases."

REVIEWERS' COMMENTS:

Reviewer #2 (Remarks to the Author):

I thank the authors for addressing my comments and limiting the functional analyses to the core genus core genomes. The additional SI Note 2 does make a convincing argument that the core genomes are indeed are a valuable proxy for the functional capacity of the identified OTUs.

The authors also mention that an SQL dump of the used primary data is available in lieu of the original database being unavailable. thus, as a final minor request, I would ask the authors to add an explicit link to this source data in the Data Availability section (even though it may be contained within the already mentioned Github repository).

Apart from this, I have no other concerns and believe that the manuscript has been improved significantly.

Reviewer #3 (Remarks to the Author):

All of my comments have been satisfactorily addressed. I thank the authors for their careful revisions.

I thank the authors for addressing my comments and limiting the functional analyses to the core genus core genomes. The additional SI Note 2 does make a convincing argument that the core genomes are indeed a valuable proxy for the functional capacity of the identified OTUs.

The authors also mention that an SQL dump of the used primary data is available in lieu of the original database being unavailable. thus, as a final minor request, I would ask the authors to add an explicit link to this source data in the Data Availability section (even though it may be contained within the already mentioned Github repository).

Apart from this, I have no other concerns and believe that the manuscript has been improved significantly.

Thanks for the constructive review process! We now include a direct link to the SQL dump as requested.

Reviewer #3 (Remarks to the Author):

All of my comments have been satisfactorily addressed. I thank the authors for their careful revisions.

Thanks for your help with improving the manuscript!